# Nonlinear transport and radio frequency rectification in BiTeBr at room temperature

Xiu Fang Lu[1,5], Cheng-Ping Zhang [2,5], Naizhou Wang [3], Dan Zhao[4], Xin Zhou [1], Weibo Gao [3], Xian Hui Chen [4], K. T. Law [2] & Kian Ping Loh [1]

Materials showing second-order nonlinear transport under time reversal symmetry can be used for Radio Frequency (RF) rectification, but practical application demands room temperature operation and sensitivity to microwatts level RF signals in the ambient. In this study, we demonstrate that BiTeBr exhibits a giant nonlinear response which persists up to 350 K. Through scaling and symmetry analysis, we show that skew scattering is the dominant mechanism. Additionally, the sign of the nonlinear response can be electrically switched by tuning the Fermi energy. Theoretical analysis suggests that the large Rashba spin-orbit interactions (SOI), which gives rise to the chirality of the Bloch electrons, provide the microscopic origin of the observed nonlinear response. Our BiTeBr rectifier is capable of rectifying radiation within the frequency range of 0.2 to 6 gigahertz at room temperature, even at extremely low power levels of −15 dBm, and without the need for external biasing. Our work highlights that materials exhibiting large Rashba SOI have the potential to exhibit nonlinear responses at room temperature, making them promising candidates for harvesting high-frequency and low-power ambient electromagnetic energy.

With the rapid proliferation of wireless technologies and portable devices, electromagnetic radiation has emerged as a promising energy source for distributed electronics[1–5]. Wireless charging and energy harvesting generally require the conversion of electromagnetic radiation into direct currents. Harvesting Wi-Fi signals between 2.4 and 5.9 GHz is challenging due to the two fundamental limitations of radiofrequency (RF) energy harvesters based on Schottky diodes[2]. Firstly, Schottky diodes require an external bias for operation due to their thermal voltage threshold. Secondly, their electron transition time limits their performance in the gigahertz frequency range[1–5]. For instance, inorganic semiconductors such as indium gallium zinc oxide (IGZO) and silicon have a cut-off frequency of around ~1 GHz[6–9], and organic semiconductors typically have very low mobility[10], restricting

their harvesting performance in the gigahertz range. Therefore, it is attractive to explore alternative mechanisms for harvesting electromagnetic energy at high frequencies without relying on external bias.

The second-order nonlinear Hall effect (NLHE), a recent addition to the Hall effect family[11–14], has emerged as a new branch of condensed matter physics research owing to its importance in probing quantum critical point[15–17] and its potential applications in energy harvesting[18–20], wireless communications and infrared detectors[1,21]. Unlike conventional or anomalous Hall effects, which require time-reversal symmetry breaking, NLHE occurs under time-reversal symmetric conditions but typically requires a Berry curvature dipole (BCD)[11,13,14]. The second-order nonlinearity contains two components: a second harmonic voltage of the driving alternating current, and a direct voltage generated

[1]Department of Chemistry, National University of Singapore, Singapore 117543, Singapore. [2]Department of Physics, Hong Kong University of Science and Technology, Hong Kong, China. [3]Division of Physics and Applied Physics, School of Physical and Mathematical Sciences, Nanyang Technological University, Singapore 637371, Singapore. [4]Department of Physics and Hefei National Laboratory for Physical Science at Microscale, University of Science and Technology of China, Hefei, Anhui 230026, P. R. China. [5]These authors contributed equally: Xiu Fang Lu, Cheng-Ping Zhang. ✉e-mail: phlaw@ust.hk; chmlohkp@nus.edu.sg

from the rectification effect[1,11,21–24]. Since the second-order nonlinearity does not depend on interfacial junctions, the rectification process is not constrained by the thermal voltage threshold or transition time inherited from a semiconductor junction[1]. However, one drawback precluding practical application is that BCD is sensitive to the band structures and the NLHE decreases rapidly as temperature increases and vanishes by room temperature[13,14,25–36], with a few exceptions reported for Weyl/Dirac semimetal TaIrTe₄[18] and BaMnSb₂[19]. In addition, BCD requires breaking both inversion symmetry and threefold rotational symmetry, which limits the range of materials for second-order nonlinear effect investigations. Therefore, exploring BCD-independent mechanisms for second-order nonlinear transport that can be robust at room temperature is important for the practical implementation of rectifiers.

On the other hand, second-order nonlinearity originating from skew scattering has been observed in threefold rotational symmetric systems, such as topological Bi₂Se₃[31]. However, these crystals do not exhibit robust room temperature response. Rashba spin-orbit interaction (SOI) and its related skew scattering are considered the microscopic origins of anomalous Hall effect, spin Hall effect, spin-orbit coupling, and other related phenomena[37–40]. In principle, Rashba materials with significant Rashba-type band splitting could generate considerable second-order nonlinear response, although the connection between Rashba splitting and the second-order nonlinear response has not been shown explicitly in experiments.

In this work, we demonstrate that a large in-plane second-order nonlinear response in a two-dimensional Rashba material, BiTeBr, that persists up to 350 K. The observed nonlinear response in BiTeBr exhibits a significant dependence on carrier density and its sign can be electrically switched by tuning the Fermi energy. The scaling relationship between the nonlinear conductivity and scattering time indicates that skew scattering is the dominant mechanism. Additionally, we demonstrate that BiTeBr can rectify wireless radiation in the Wi-Fi band (2.4 GHz and 5.9 GHz) without an external bias. These findings suggest that 2D BiTeBr rectifiers can have broadband operation.

## Results and discussion

The crystal structure of BiTeBr is illustrated in Fig. 1a, b from a side and top view, respectively. The material possesses electric polarization along the c-axis, while Berry curvature dipole is forbidden in the *ab*-plane due to the threefold rotational symmetry. This allows us to

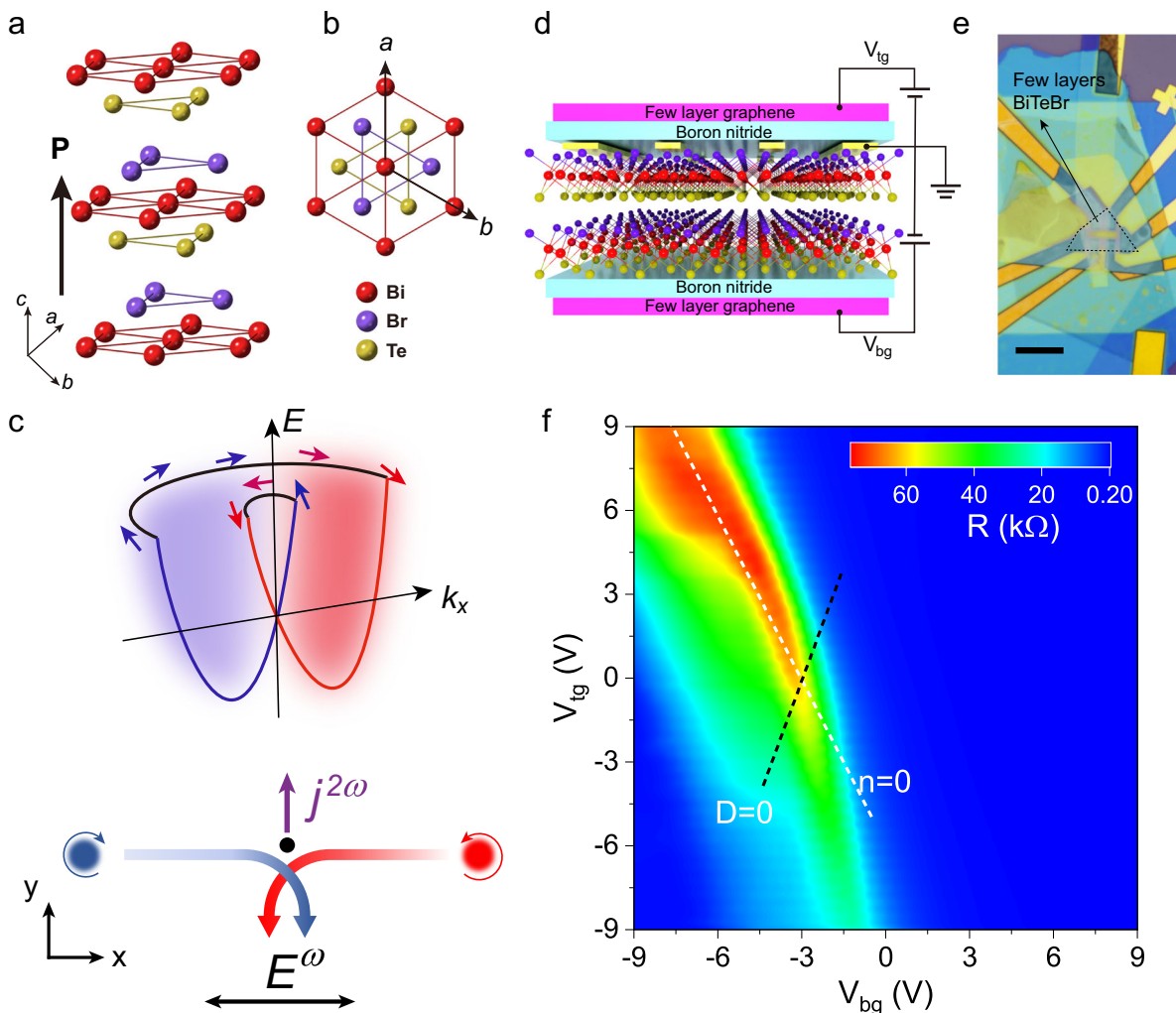

**Fig. 1 | Crystal structure and basic characterization of BiTeBr. a, b** Crystal structure of BiTeBr from side view (**a**) and top view (**b**). Br, Bi, and Te layers are alternately stacked along the c axis, breaking the mirror symmetry and resulting in a polarization axis along the c direction. **c** Schematic figure shows skew scattering arising from Rashba band splitting contributing to nonlinear transport response. **d** The schematic geometry of the h-BN encapsulated, dual gated, few layers BiTeBr device. **e** Optical image of our BiTeBr device. The dashed black line traces the encapsulated few layers BiTeBr flake. Scale bar, 10 μm. **f** The four-probe longitudinal resistance R (color scale) as a function of top gate voltage ($V_{tg}$) and bottom gate voltage ($V_{bg}$) at T = 50 K. The dashed white and black line represent the charge neutrality (n = 0) and zero displacement field (D = 0), respectively.

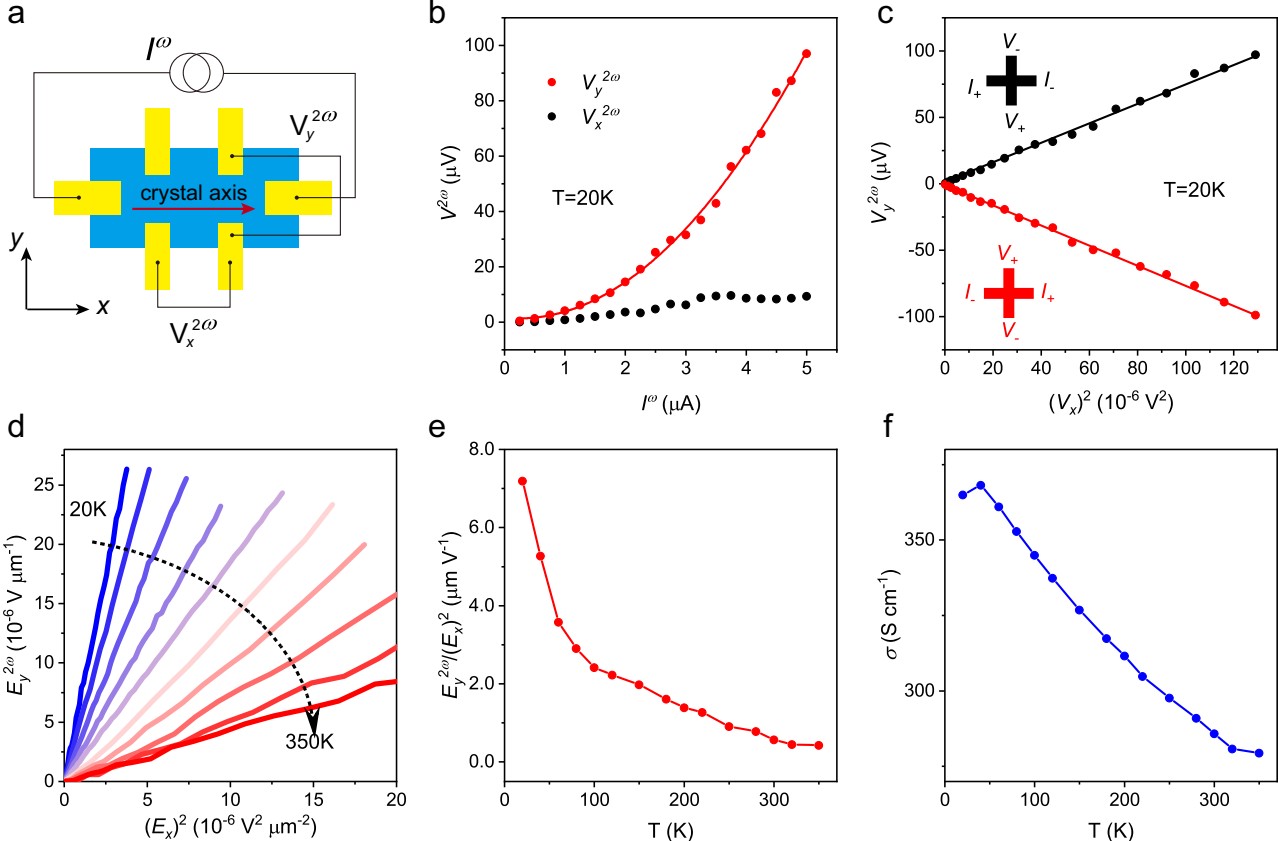

**Fig. 2 | Nonlinear response measured in BiTeBr device. a** Schematic illustration of the nonlinear electrical transport measurement in BiTeBr Hall bar device. Current is applied along the crystal axis with a frequency $\omega$, and the measured transverse ($V_y^{2\omega}$) and longitudinal voltage ($V_x^{2\omega}$) at a frequency of $2\omega$. **b** Second harmonic transverse ($V_y^{2\omega}$) and longitudinal voltage ($V_x^{2\omega}$) as a function of alternating current with frequency $\omega$ in BiTeBr device at 20 K. The thickness of the device is ~ 4 nm.

**c** $V_y^{2\omega}$ depends linearly on the square of longitudinal voltage $V_x$ and changes the sign when the current direction and voltage probe electrodes are simultaneously reversed. **d** $E_y^{2\omega}$ dependent of $(E_x)^2$ measured at temperature ranging from 20 K to 350 K. **e** The nonlinear susceptibility $E_y^{2\omega}/(E_x)^2$ as a function of temperature. **f** Temperature dependence of conductivity $\sigma$ of this BiTeBr device.

investigate extrinsic scattering contribution to the second-order nonlinearity. For example, skew scattering arises from the asymmetry of the differential scattering cross section due to the inherent chirality of the Bloch electrons, which is induced by the inversion symmetry breaking and spin-orbit coupling within the material. As depicted in Fig. 1c, the Rashba spin-orbit coupling in these systems leads to the splitting of electronic bands and generates chiral Bloch electrons. When the self-rotating wave-packet is scattered by an impurity centre, the asymmetry of the scattering cross section will give rise to a rectified nonlinear Hall voltage[1,25,31–33].

To study the second-order nonlinearity generated by skew scattering, we fabricated high-quality, dual-gated few-layer BiTeBr device encapsulated in h-BN, as presented in Fig. 1d, e. We employed second harmonic generation (SHG) and scanning transmission electron microscopy (STEM) techniques to determine the crystal axes of exfoliated BiTeBr flakes. The Supplementary Fig. S2 contains details on the orientation of the BiTeBr flakes. After confirming the crystal axes orientation, we aligned the current electrodes along the crystal axis. The resistance as a function of the top gate ($V_{tg}$) and bottom gate ($V_{bg}$) is displayed in Fig. 1f, featuring the charge neutrality line ($n = 0$, marked with a white dashed line) and the zero electrical displacement field ($D = 0$, marked with a black dashed line).

We investigate the nonlinear transport in BiTeBr by driving a harmonic current ($I^\omega$) along the crystal axis of BiTeBr crystal with a fixed frequency ($\omega = 17.777$ Hz), and measure the voltage at second harmonic frequencies ($2\omega = 35.554$ Hz) in both longitudinal and transverse directions, as shown in Fig. 2a. In our experiment, the

$x$-axis is defined as the current direction and the $y$-axis denotes the transverse direction to the current. We first examined the nonlinear response in a 4 nm-thick BiTeBr device, as shown in Fig. 2. At $T = 20$ K, the second harmonic transverse voltage $V_y^{2\omega}$ demonstrates a clear quadratic relationship with the current under zero magnetic field and is significantly larger than the longitudinal second harmonic voltage $V_x^{2\omega}$ (Fig. 2b). $V_y^{2\omega}$ scales linearly with the square of the fundamental longitudinal voltage $V_x$, and its sign changes when simultaneously reversing the current direction and voltage probe electrodes (Fig. 2c). We also measured $V_y^{2\omega}$ as a function of temperature up to 350 K, as illustrated in Fig. 2d, where $E_y^{2\omega}$ and $E_x$ represent the second harmonic transverse and first harmonic longitudinal electric fields, respectively. Here, $E_y^{2\omega} \equiv V_y^{2\omega}/L_y$, and $E_x \equiv V_x/L_x$, where $L_y$ and $L_x$ are the transverse and longitudinal lengths of the Hall bar device. $E_y^{2\omega}$ exhibits a linear dependence on $(E_x)^2$ at all temperatures, with the slope of $E_y^{2\omega} \cdot (E_x)^2$ decreasing monotonically with increasing temperature. Nevertheless, $E_y^{2\omega}$ can persist up to 350 K, and the nonlinear Hall susceptibility[14] ($E_y^{2\omega}/(E_x)^2$), which characterizes the nonlinear response, can reach ~ 0.57 $\mu$m/V at 300 K. This nonlinear susceptibility is relatively high among 2D materials exhibiting nonlinear response[13,14,18–36], thus offering immense potential for RF rectification. The temperature dependence of $E_y^{2\omega}/(E_x)^2$ and conductivity ($\sigma$) of the device are shown in Fig. 2e, f, both of which exhibit similar behavior.

To explore the physical origin of the transverse second harmonic response, we systematically measure the $V_y^{2\omega}$ as function of top ($V_{tg}$) and bottom ($V_{bg}$) gate voltages using a dual-gate device. The double

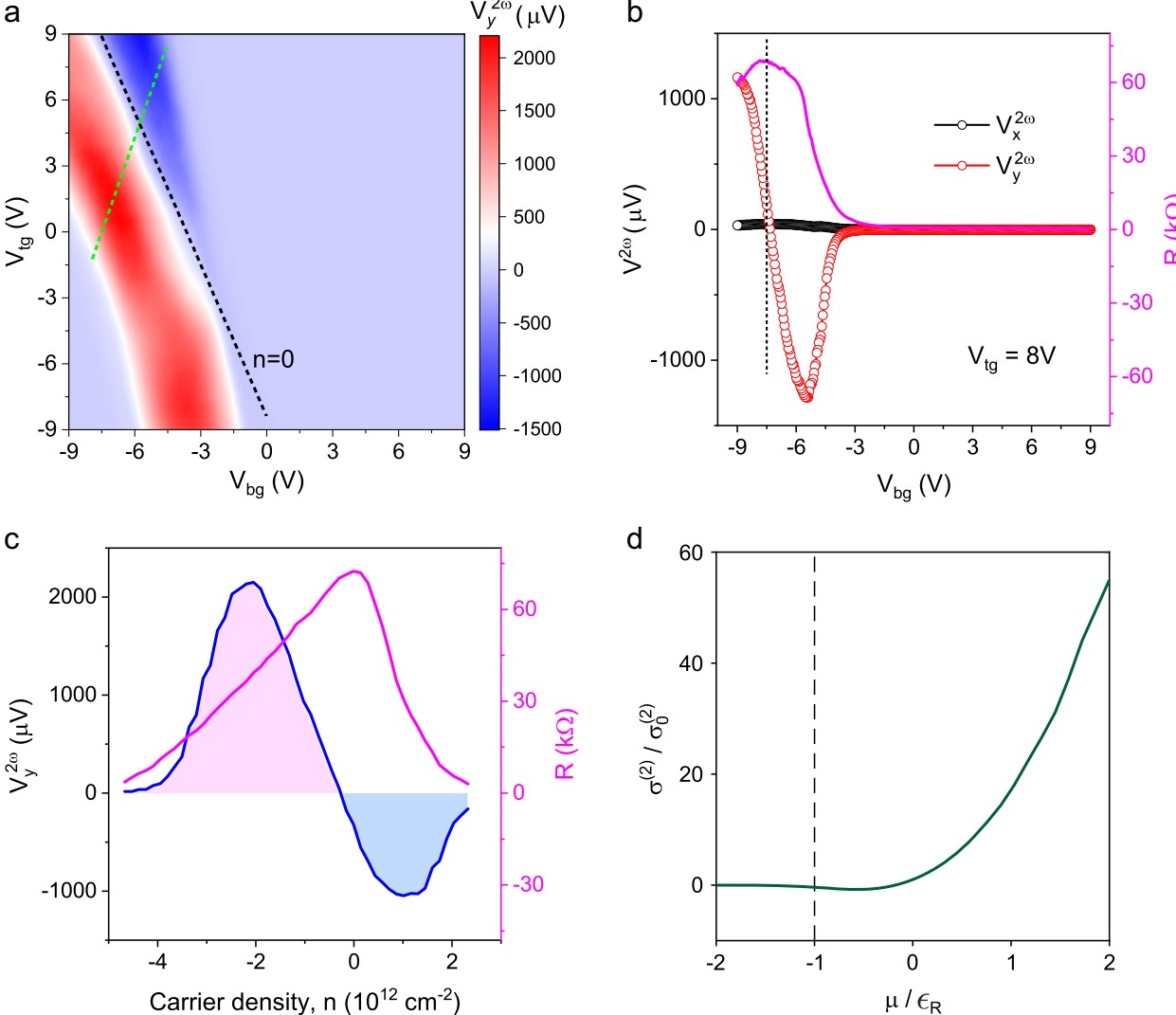

**Fig. 3 | The nonlinear response in dual gated few layers BiTeBr device. a** Second harmonic transverse voltage $V_y^{2\omega}$ (color scale) as a function of top and bottom gate $V_{tg}$ and $V_{bg}$ at 50 K. An AC current $I^\omega = 2\,\mu$A with frequency 17.777 Hz is applied along the crystal axis. The dashed black line ($n = 0$) corresponds to the maximum resistance (charge neutrality in Fig. 1f). The dashed green line denotes a trajectory along which $n$ is varied with constant $D$. **b** $V_y^{2\omega}$ (red circles) and $V_x^{2\omega}$ (black circles) as a function of $V_{bg}$ with $V_{tg} = 8$ V are shown at left. The resistance $R$ (pink line) as a function of $V_{bg}$ is shown at right. **c** $V_y^{2\omega}$ (left blue line) and $R$ (right, pink line) as a function of carrier density $n$ along the dashed green line in (**a**). **d** Calculated gate dependence of the nonlinear conductivity, where the black dashed line indicates the band edge. The parameter $\lambda k_0^3/\epsilon_R = 0.1$ is used for the skew scattering calculation, with $\epsilon_R = \frac{\alpha_R^2}{4t}$ and $k_0 = \frac{\alpha_R}{2t}$. The nonlinear conductivity is normalized against values at zero chemical potential.

gated device structure allows us to independently control the carrier concentration $n$ and electric displacement field $D$[13] to investigate their influence on nonlinear transport in BiTeBr. Similar studies have been carried out on nonlinear Hall effect in Berry curvature dipole-driven systems[13]. The $V_y^{2\omega}$ as a function of top and bottom gate voltages at 50 K is shown in Fig. 3a, where $V_y^{2\omega}$ shows a strong dependence on carrier density $n$. The dashed black line in Fig. 3a denotes the charge neutrality line ($n = 0$) in Fig. 1f, while the dashed green line represents a trajectory at constant $D$ when $n$ varies. We observe an obvious sign reversal of $V_y^{2\omega}$ around $n = 0$, that appears to be independent of $D$. In Fig. 3b, we show the $V_y^{2\omega}$ and $V_x^{2\omega}$ as a function of $V_{bg}$ at a fixed $V_{tg} = 8V$ on the left axis, with the resistance of the same device showing on the right axis. $V_y^{2\omega}$ reverses sign with the decreasing $V_{bg}$. This sign reversal occurs around the maximum of the resistance, corresponding to the charge neutrality point. The sign reversal of $V_y^{2\omega}$ around $n = 0$ suggests that the second-order nonlinear response of BiTeBr is related to the band structure and shows a strong dependence on band filling. In

Fig. 3c, we show the carrier density $n$-dependent $V_y^{2\omega}$ (left) and resistance (right) along the dashed green line (with a constant $D$) in Fig. 3a. The $V_y^{2\omega}$ also reverses its sign around the maximum of the resistance and exhibits a bipolar behavior as a function of $n$. The calculated gate dependence of the nonlinear conductivity is shown in Fig. 3d, where the black dashed line indicates the band edge. There is a rapid growth of nonlinear conductivity when the chemical potential is gated from the band edge, which explains the rapid change in the nonlinear transverse response with gate voltage when the Fermi level is tuned from the charge neutrality point.

To further investigate the microscopic origins of our observed nonlinear transport, we study the scaling behavior of the nonlinear response as a function of temperature for BiTeBr with various thicknesses. $E_y^{2\omega}$ depends linearly on $(E_x)^2$ over the entire temperature range, and the nonlinear susceptibility $E_y^{2\omega}/(E_x)^2$ and conductivity $\sigma$ show similar temperature-dependent behavior in all BiTeBr devices (Supplementary Figs. S7 and S8). As shown in Fig. 4,

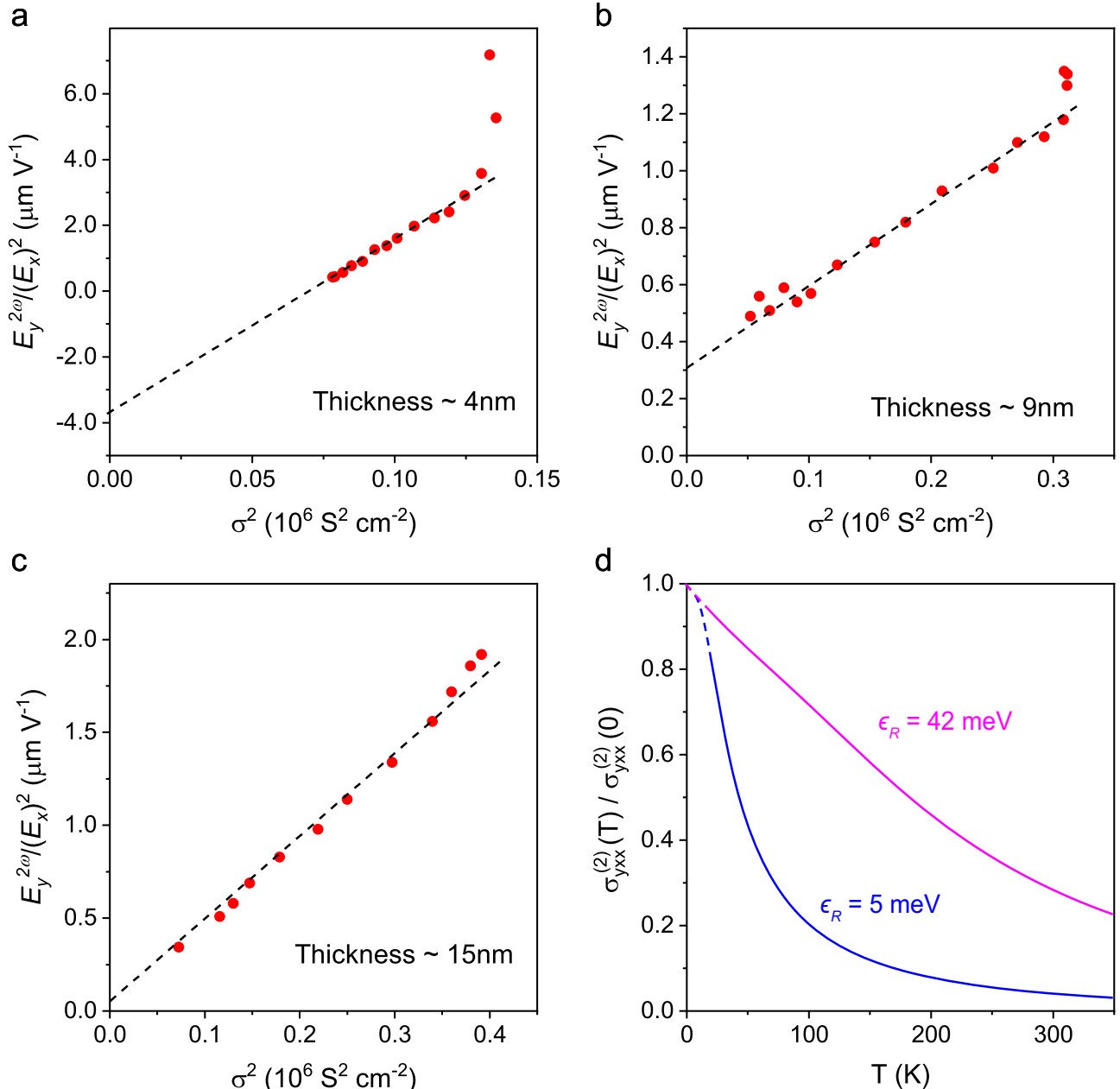

**Fig. 4 | Temperature dependence of nonlinear susceptibility in BiTeBr. a–c** The nonlinear susceptibility ($E_y^{2\omega}/(E_x)^2$) as a function of the square of conductivity ($\sigma^2$) for BiTeBr devices with 4 nm thickness (**a**), 9 nm thickness (**b**), and 15 nm thickness (**c**). **d** The calculated nonlinear conductivity $\sigma_{yxx}^{(2)}$ (normalized by the zero temperature value) as a function of temperature for Rashba band splitting $\epsilon_R = 42$meV (pink) and $\epsilon_R = 5$meV (blue). The calculation incorporates the temperature dependence of the scattering time for $T > 20$K, and extends to temperatures below 20 K, as indicated by the dashed lines.

the nonlinear susceptibility scales linearly with $\sigma^2$ and is consistent with the relation:

$$\frac{E_y^{2\omega}}{(E_x)^2} = \xi\sigma^2 + \eta, \tag{1}$$

where $\xi$ and $\eta$ are constants, except for deviations of a few low temperature points which might be influenced by disorder[27,33]. Since $\sigma = J/E$, where **J** is the current density, we can write second-order current density as: $J_y^{(2)} = \sigma_{yxx}^{(2)}(E_x)^2 = \sigma E_y^{(2)}$, $\frac{E_y^{(2)}}{(E_x)^2} = \frac{\sigma_{yxx}^{(2)}}{\sigma} = \xi\sigma^2 + \eta$. Thus, the second-order conductivity $\sigma_{yxx}^{(2)}$ scales as $\sigma^3$ and $\sigma$ with the coefficients $\xi$ and $\eta$, respectively. Since $\sigma$ is linearly dependent on the scattering

time $\tau$, the $\sigma_{yxx}^{(2)}$ has two contributions that scale as $\tau^3$ and $\tau$, respectively[11,14,31]. The slope $\xi$ of Eq. (1) quantifies the contribution of $\tau^3$ in $\sigma_{yxx}^{(2)}$, which originates from skew scattering. The intercept $\eta$ of Eq. (1) quantifies the contribution of $\tau$ in $\sigma_{yxx}^{(2)}$, which has contributions from Berry curvature dipole and side jumps[14,31,33]. In our BiTeBr case, the Berry curvature dipole is forbidden due to the threefold rotational symmetry. Thus, we ascribe the $\tau$-linear contribution to the side jump mechanism. Therefore, the two main contributors to the nonlinear response in BiTeBr are skew scattering and side jump.

To have an idea on the influence of sample thickness on magnitude of skew scattering, we compare $\xi$, the slope of the nonlinear susceptibility ($E_y^{2\omega}/(E_x)^2$) versus $\sigma^2$, for devices with different thick-

nesses of BiTeBr, as shown in Fig. 4. The carrier densities $n$ for 4 nm, 9 nm, and 15 nm thick devices are $2.6 \times 10^{19}\,\mathrm{cm^{-3}}$, $2.5 \times 10^{19}\,\mathrm{cm^{-3}}$, and $2.8 \times 10^{19}\,\mathrm{cm^{-3}}$, respectively. Since the change in carrier density is less than 10% over the temperature range of 10 K to 300 K, and that these carrier densities are all located far away from $n = 0$ in the $V_y^{2\omega}$ map in Fig. 3a, we can compare how $E_y^{2\omega}/(E_x)^2$ scales with $\sigma^2$ without considering the effect due to different $n$. The slope $\xi$, derived from the nonlinear susceptibility versus $\sigma^2$ plot, is $6.9 \times 10^{-15}\,\mathrm{m^3 V^{-1} S^{-2}}$ for the 4 nm-thick device, which is larger than the 9 nm-thick device ($3.0 \times 10^{-16}$ $\mathrm{m^3 V^{-1} S^{-2}}$) and 15 nm-thick device ($4.5 \times 10^{-16}\,\mathrm{m^3 V^{-1} S^{-2}}$). The enhancement of skew scattering in thinner device might be due to the enhanced surface Rashba spin-splitting in BiTeBr crystal[41–44]. The slope $\xi$ of 15 nm device is larger than 9 nm device, indicating that surface state is not the only source for skew scattering. The contribution of bulk Rashba spin-splitting is still remarkable, especially in thicker device. The sign of $\eta$, which indicate the contribution from side jump, changes from negative to positive with the increase of thickness. This is similar to the sign change of side jump contribution observed in the anomalous Hall effect[45]. The scaling dependence of $\sigma^3$ and $\sigma$ with respect to skew scattering and side jump contributions to $\sigma_{yxx}^{(2)}$ are analyzed in detail in Supplementary Fig. S9, the result shows that skew scattering dominates the nonlinear second order response.

To further prove the nonlinear response observed is arising from skew scattering and complying with the threefold rotational symmetry of BiTeBr[1], we investigate the angle- dependent nonlinear response. The nonlinear longitudinal and transverse response are measured in a disc-shape device with different current injection angle, and the results are shown in Supplementary Figs. S10 and S11. Both the longitudinal and transverse nonlinear response exhibit a threefold symmetry, fully consistent with the threefold symmetry of BiTeBr and suggest that the skew scattering is the dominant mechanism.

In the following, we provide a microscopic origin of the large nonlinear response in BiTeBr. The BiTeBr crystals belong to the $C_{3v}$ point group symmetry, which forces the Berry curvature dipole to vanish. However, skew scattering only requires inversion symmetry breaking and can contribute to nonlinear response. Under $C_{3v}$ point group symmetry, the nonzero nonlinear conductivity components are $\sigma_{yxx}^{(2)} = \sigma_{xxy}^{(2)} = \sigma_{xyx}^{(2)} = -\sigma_{yyy}^{(2)}$. When the current is applied along the $x$ direction, the nonlinear Hall voltage along the $y$ direction is determined by $\sigma_{yxx}^{(2)}$, therefore we will focus on this component in the following.

We describe the BiTeBr thin film with the 2D Rashba Hamiltonian

$$\hat{H}_0 = \epsilon_0(k) + \alpha_R \left( k_y \sigma_x - k_x \sigma_y \right) + \frac{\lambda}{2} \left( k_+^3 + k_-^3 \right) \sigma_z, \tag{2}$$

where $\epsilon_0(k) = t k^2$ is kinetic energy, $\alpha_R$ characterize the Rashba spin-orbit coupling, and we have added a third-order warping term. For simplicity, we first ignore the kinetic energy. With the semiclassical Boltzmann equation, we obtain the second-order conductivity from skew scattering as:

$$\sigma^{(2,skew)} = \pm \mathrm{sgn}(\lambda) \frac{e^3 \alpha_R \tau^3}{\hbar^3 \tilde{\tau}}, \tag{3}$$

where the "$\pm$" symbol represents the Rashba bands, $\mathrm{sgn}(\lambda)$ represents the sign function of the third-order warping strength, $\tau \approx \frac{\hbar V_F k_F}{\pi^2 n_i \alpha^2 \alpha_R^2}$ is the isotropic scattering time and $\tilde{\tau} \approx 16/(\pi^2 c_1 n_i \alpha^3 |\lambda| k_F)$ is the skew scattering time. Here, $V_F$ is the Fermi velocity, $k_F$ is the Fermi wave vector, $n_i$ is the impurity density, $\alpha$ characterizes the scattering strength and $c_1$ is a numeric constant. Based on the above formalism, we can rationalise several features about the nonlinear response in BiTeBr. (1) The nonlinear conductivity of the two Rashba bands carries opposite signs. This is general and well-expected. In the limit of zero spin-orbit coupling, the two Rashba bands will be degenerate and the

contribution from the two bands should cancel each other. Therefore, a sign change will be expected when the Fermi energy is tuned across the bands. (2) The second-order conductivity exhibits a $\tau^3$ relationship, which agrees with our experimental scaling behavior. (3) The scattering time $\tau \propto k_F$ and the skew scattering time $\tilde{\tau} \propto k_F^{-1}$, which implies that the nonlinear conductivity $\sigma^{(2)} \propto k_F^4$. This explains the rapid change in the non-linear Hall response with back gate voltage when we tune the Fermi level from the charge neutrality point in Fig. 3b. Next, we study the temperature dependence of the nonlinear response. At finite temperature $T$, the nonlinear conductivity is calculated as

$$\sigma_{yxx}^{(2)}(T) \simeq \pm \mathrm{sgn}(\lambda) \frac{e^3 \alpha_R \tau^3}{\hbar^3 \tilde{\tau}} \tanh \frac{\alpha_R k_F}{2 k_B T} \tag{4}$$

The scattering time is inversely proportional to the resistance, which shows a linear dependence on the temperature in the experiment with $T > 20$K. Therefore, the temperature dependence of the scattering time can be approximated as $\tau^{-1} = \tau_0^{-1}(1 + aT)$ with $a \approx 1.1 \times 10^{-3}\,\mathrm{K^{-1}}$ fitted from Fig. 2f. When the Fermi energy is near the band edge, we can estimate $\alpha_R k_F \approx \epsilon_R$, therefore we get the temperature dependence of the second-order conductivity

$$\frac{\sigma_{yxx}^{(2)}(T)}{\sigma_{yxx}^{(2)}(0)} \simeq (1 + aT)^{-3} \tanh \frac{\epsilon_R}{2 k_B T} \tag{5}$$

When $k_B T \gg \epsilon_R$, $\tanh \frac{\epsilon_R}{2k_B T} \to 0$, causing the nonlinear conductivity to vanish. Therefore, a large Rashba spin-orbit coupling is crucial to enable the nonlinear response to survive at room temperature, and this is the crucial finding of this work. To illustrate the important role of the Rashba spin-orbit coupling, we plot the temperature dependence in Fig. 4c with both $\epsilon_R = 42$ meV (the Rashba splitting energy in BiTeBr)[41–44] and a smaller $\epsilon_R = 5$ meV. It is evident that the nonlinear response will decay very fast with temperature rise if the Rashba spin-orbit coupling is small ($\epsilon_R = 5$ meV), but can survive to room temperature in BiTeBr with $\epsilon_R = 42$ meV. We also employ Eq. (5) to fit our experimental results, as presented in Supplementary Fig. S12. The good agreement between the theoretical equation and experimental data further support that a strong Rashba spin-orbit coupling is responsible for the room temperature nonlinear response observed in BiTeBr.

Table 1 provides a comparison of the second-order nonlinear response of various materials. Our BiTeBr device generates a 100 μV nonlinear voltage with only 5 μA input current, which is larger than most previously reported materials. This value is comparable to the hBN/graphene/hBN moiré superlattice[32] and is surpassed only by the twisted bilayer WSe$_2$[34]. Although materials such as TaIrTe$_4$[18] and BaMnSb$_2$[19] have been reported to exhibit room temperature nonlinear Hall effect, the nonlinearity response in those systems is significantly smaller than that of our BiTeBr device. This strong room temperature nonlinearity suggests that BiTeBr can be used as a RF harvester in distributed electronics to rectify RF signals into DC voltage.

Finally, we demonstrate room-temperature wireless RF rectification without any external bias using BiTeBr as a rectifier. The experimental setup for measuring rectification in the BiTeBr device is depicted in Fig. 5a, where the device is exposed to electromagnetic waves spanning a frequency range of 0.2 to 6.0 GHz. Remarkably, the BiTeBr device exhibits excellent rectification across this wide RF frequency range, and its rectification performance scales linearly with the power of the RF source, as shown in Fig. 5b. These findings indicate that the second harmonic nonlinear response of the BiTeBr device can be effectively utilized as a microwave energy harvester with a broad bandwidth. By analyzing the logarithmic output DC electrical voltage of our BiTeBr device, we observe that rectification begins at approximately −15 dBm power (-0.03 mW), as depicted in Fig. 5c. This power

**Table 1 | The second-order nonlinear response in different materials**

| Materials | Highest working temperature (K) | Input current maximum (µA) | Output voltage maximum (µV) | References |
|---|---|---|---|---|
| WTe$_2$ (bilayer) | 100 | 1 | 5 | ref. 13 |
| WTe$_2$ (few-layer) | 100 | 600 | 30 | ref. 14 |
| TaIrTe$_4$ | 300 | 600 | 120 | ref. 18 |
| Bi$_2$Se$_3$ surface | 200 | 1500 | 20 | ref. 31 |
| hBN/graphene/hBN moiré superlattice | 220 | 5 | 100 | ref. 32 |
| Twisted bilayer WSe$_2$ | 30 | 0.04 | 20,000 | ref. 34 |
| Strained WSe$_2$ (monolayer) | 140 | 4.5 | 20 | ref. 35 |
| T$_d$-MoTe$_2$ (in-plane) | 100 | 97 | 125 | ref. 26 |
| T$_d$-MoTe$_2$ (c-axis) | 40 | 5000 | 40 | ref. 27 |
| Corrugated bilayer graphene | 15 | 0.1 | 2 | ref. 36 |
| Ce$_3$Bi$_4$Pd$_3$ | 4 | 10,000 | 0.8 | ref. 28 |
| α-(BEDT-TTF)$_2$I$_3$ | 4.2 | 1000 | 9 | ref. 30 |
| BaMnSb$_2$ | 400 K | 100 | 250 | ref. 19 |
| Focused ion beam deposited Pt | 375 K | 50 | 120 | ref. 20 |
| BiTeBr | > 350 | 5 | 100 | This work |

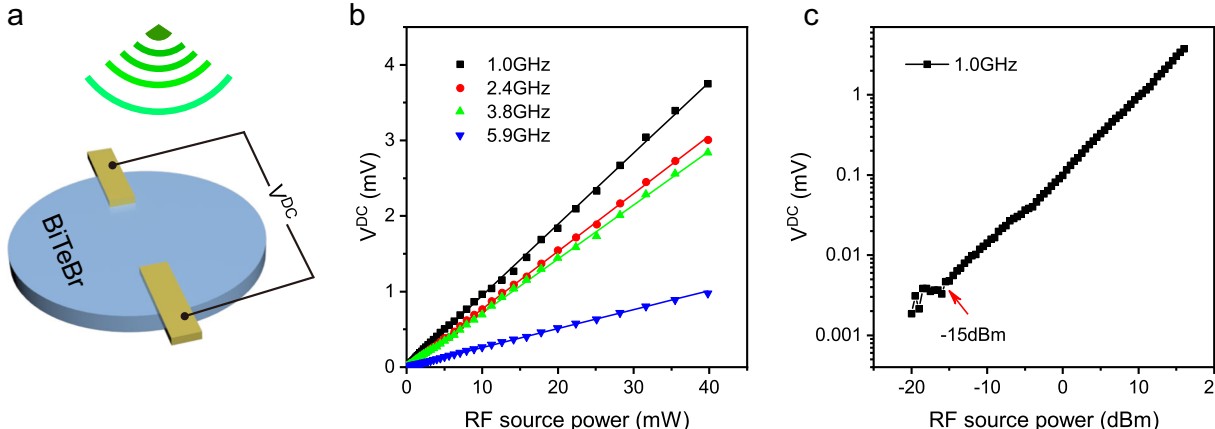

**Fig. 5 | Wireless RF rectification measured on a 22nm-thick BiTeBr device at room temperature. a** Schematic of the rectification measurement on BiTeBr device. Antennas with transmitting frequencies ranging from 0.2 to 6 GHz were used to receive microwaves from the signal generator and apply wireless RF to the BiTeBr device. The antenna was aligned along the crystal axis of BiTeBr and the output DC electrical voltage was measured in the direction perpendicular to the crystal axis. **b** The rectified DC electrical voltage as a function of the wireless radiofrequency power under frequencies of 1.0, 2.4, 3.8, and 5.9 GHz. **c** The logarithmic output DC electrical voltage as a function of RF power with dBm unit.

level approaches the ambient RF power range (−20 to −10 dBm)[46]. There is much room for further enhancing the rectification performance of the BiTeBr device through appropriate antenna design and impedance matching circuits. The ability of BiTeBr to detect and rectify low-power wireless RF signals at high frequencies positions it as a promising candidate for low-power ambient electromagnetic energy harvesting.

In summary, we have demonstrated a large second harmonic nonlinear response in BiTeBr, a Rashba material. The skew scattering associated with large Rashba spin splitting ($E_R > K_B T$) provides a straightforward approach for selecting materials exhibiting large nonlinear electric responses that persist above room temperature. This principle can be applied to a wide range of Rashba materials that possess chiral Bloch electrons, as well as two-dimensional heterostructures with engineered Rashba interfaces, thereby expanding the scope of materials that exhibit nonlinear response at room temperature. The opposite sign of the nonlinear conductivity of the two Rashba bands allows for electrical modulation of the nonlinear response. Our BiTeBr rectifier rectifies wireless RF signals over 0.2 to 6 GHz and works at low power RF signal at −15 dBm under zero external bias. Our discoveries have significant implications for the utilization of Rashba-type nonlinear responses in high-frequency, low-power RF rectification applications, and paves the way for advancements in wireless communication technologies.

## Methods

### Sample and device fabrication

High-quality bulk BiTeBr single crystals were grown by the flux method. Two methods were used to exfoliate BiTeBr flakes: direct mechanical exfoliation and Al$_2$O$_3$-assisted mechanical exfoliation[47]. The dual-gated few-layer BiTeBr devices were fabricated by the layer-by-layer dry transfer method which consists of two stages. The first stage was conducted under ambient conditions. Hexagonal boron nitride (hBN) flakes (thickness ~ 20 nm) were exfoliated on SiO$_2$/Si substrate, then picked up using a polymer-based dry transfer method and placed onto a prepared few-layer graphene flake. 350 °C-post annealing was conducted on this hBN/graphene heterostructure under H$_2$/Ar gas flow for 8 hours to get a clean hBN upper surface. The second stage was performed fully inside an Argon-filled glovebox with O$_2$ and H$_2$O < 0.1 ppm. Few-layer BiTeBr flakes were exfoliated from bulk crystal using purified PDMS film and transferred onto the annealed hBN/graphene substrate. The transfer process was performed without

heating to avoid leaving polymer residue. The crystalline axis of the transferred flakes was then determined using second harmonic generation (SHG) measurement after coating with a PMMA polymer layer in the glovebox. Using standard electro-beam lithography method, Ti/Au contacts were deposited onto the BiTeBr/hBN/graphene structure with the current electrodes aligned along the crystal axis of BiTeBr flakes. Few-layer graphene, hBN flakes (thickness ~ 20 nm) were sequentially picked up and then transferred onto the electrodes/BiTeBr/hBN/graphene structure. In this BiTeBr dual-gated device, the carrier density n can be obtained from $n = \frac{\varepsilon_0 \varepsilon^{hBN}}{e}(\frac{V_{tg}-V_{tg0}}{h_{tg}} + \frac{V_{bg}-V_{bg0}}{h_{bg}})$, where $V_{tg}$ and $V_{bg}$ are the top and bottom gate voltage, $h_{tg}$ and $h_{bg}$ are the thickness of the top and bottom hBN, $\varepsilon_0 = 8.85 \times 10^{-12} Fm^{-1}$ is the vacuum permittivity, $\varepsilon^{hBN} \approx 3$ is the relative dielectric constant of hBN. The electric displacement field D can be obtained from $D = \frac{\varepsilon^{hBN}}{2}(\frac{V_{tg}-V_{tg0}}{h_{tg}} - \frac{V_{bg}-V_{bg0}}{h_{bg}})$.

### Nonlinear transport measurement

Transport measurements were carried out in a Lake Shore Cryostat (CRX-VF probe station). A harmonic current was applied to the BiTeBr device by Keithley 6221 current sources. First- and second-harmonic voltage signals along both longitudinal and transverse directions were collected with lock-in amplifiers (Stanford Research System SR830). The phase of the first- and second-harmonic voltage is confirmed to be ~0° and 90° during the measurement. The top and bottom gate voltage $V_{tg}$ and $V_{bg}$ were applied through Keithley 2636B source meters.

### RF rectification measurement

A Stanford Research Systems SG386 signal generator was used to generate RF signals. Antennas with transmitting frequencies ranging from 0.2 to 6 GHz were used to receive microwaves from the signal generator and apply wireless RF signal to the BiTeBr device. A Keithley 2182 nanovoltmeter was used to measure the output rectified DC voltage.

## Data availability

The source data generated in this study are provided in the Source Data file and are also available from the corresponding author upon reasonable request. All other data are available from the corresponding author upon request. Source data are provided with this paper.

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

## Acknowledgements

K.P.L. acknowledges the support from Singapore's National Research Foundation, Prime Minister's Office, Singapore under Competitive Research Program Award NRF-CRP22-2019-0006. K.T.L. acknowledges the support of the Ministry of Science and Technology of China and the HKRGC through Grants No. 2020YFA0309600, No. RFS2021-6S03, No. C6025-19G, No. AoE/P-701/20, No. 16310520, No. 16310219, and No. 16307622. W.G. acknowledges the financial support from the Singapore National Research Foundation through its Competitive Research Program (CRP Award No. NRF-CRP22-2019-0004).

## Author contributions

K.P.L. and K.T.L. conceived and supervised the project. X.F.L. fabricated the devices, performed the transport and wireless RF rectification measurement with help from N.W. and W.G. X.F.L. and K.P.L. analyzed the data. C.P.Z. and K.T.L. performed the theoretical calculations. X.Z. performed the STEM measurement. N.W. performed the SHG measurement. D.Z. and X.H.C. grew the BiTeBr single crystals. X.F.L, C.P.Z., K.T.L., and K.P.L. wrote the manuscript, with input from all authors.

## Competing interests

The authors declare no competing interests.
