## [Peer Review File · Nature Communications]

Reviewers' Comments:

Reviewer #1:

Remarks to the Author:

The paper by Zan et al. presents experimental studies of the second-order transport on 2D BiTeBr Rashba systems. Recently, the second order transport of noncentrosymmetric materials have attracted significant interest. The results present in this paper are definitely of interest. However, before I can support publication, the following issues need to be resolved:

1. The authors showed that the transverse nonlinear voltage V_{yxx}^2 is nonzero (i.e., injecting current along x and measuring the nonlinear voltage along y). Can the authors do the same experiments but rotate by 90 deg? V_{xyy} should be zero whereas V_{yyy} should be nonzero (with opposite sign). This is important to rule out various extrinsic second order effects (e.g. schottky diodes at the sample/contact junction, etc).
2. About the wireless rectification, Fig. 5a presents a device schematic that is cross-shaped, which is clearly different from the actual device picture in Fig. 1e. Are the authors using the same device shown in Fig. 1e to do wireless rectification or different device? More importantly, how do the authors control the electric field of the microwave and how do the authors know that the observed effect arises from a transverse second order response? In Ref. 18, the authors used antenna to control the direction of the incident microwave electric field. That would be useful if the authors want to claim that the rectification arises from the transverse nonlinearity.
3. Interpretation of the data: In my opinion, what the authors observed is a second-order transverse signal but NOT a second-order Hall signal. Specifically, a Hall effect should be the antisymmetric component of the transverse signal, meaning that $\sigma_{yxx} = -\sigma_{xyx}$. By contrast, as the authors correctly pointed out in their own paper, the $C3v$ group of BiTeBr requires their nonlinear effect to be symmetric, i.e., $\sigma_{yxx} = \sigma_{xyx}$. In fact, the authors cited the theory work of Ref. 1 to support their experimental data (correctly so because Ref. 1 discussed the symmetric nonlinear effect in $C3$ -symmetric systems, just like this experimental work). However, Ref. 1 has NEVER claimed that what they predicted is a Hall effect (they only call it a second order nonlinear effect). Therefore, I think the authors should change their claim from nonlinear Hall effect to nonlinear transverse transport. I recommend the authors to read the theory paper by Tsirkin and Souza (<https://arxiv.org/pdf/2106.06522.pdf>), which has a great pedagogical description of what can be interpreted as a Hall effect (being transverse is necessary but insufficient).

Reviewer #3:

Remarks to the Author:

The nonlinear Hall effect is expected to be a new route for high-frequency energy harvesting. However, the lower working temperature of most previously reported nonlinear Hall materials significantly limits its applications. In this manuscript, X. F. Lu et al. reported a room temperature nonlinear Hall effect in few-layer BiTeBr. A notable conclusion from their scaling analysis is that skew scatterings are the primary mechanism of the observed nonlinear Hall effect, which has traditionally been observed only at low temperatures. According to their theoretical analyses, the robustness of the nonlinear Hall effect in their devices is due to the strong Rashba spin-orbit coupling. They also demonstrated harvesting of high-frequency and low-power ambient electromagnetic energy with their prototype devices.

I believe their results are interesting and important and can be considered for publication in Nature Communications. However, there are still certain concerns that need to be addressed prior to the acceptance of this manuscript.

1. As mentioned in the manuscript, under $C3v$ symmetry, $\sigma_{\{yxx\}}$ could be nonzero, while $\sigma_{\{xyy\}}$ should always be zero. Can the authors demonstrate this concept with their existing devices?
2. As mentioned in the manuscript, for the $C3v$ symmetry, $\sigma_{\{yyy\}} = -\sigma_{\{yxx\}}$, the longitudinal second harmonic response should be observed when the current is applied along the y direction (perpendicular to the crystal axis). Such longitudinal second-order responses have

already been reported in hBN/graphene moiré superlattice (ref. 32) and MnBi₂Te₄. Could the authors illustrate this in their experiment?

3. Can the authors show the data measured between the other pairs of contacts in the 4nm and 9nm device? Are they in agreement with the demonstrated data?

4. What's the current-voltage characteristic of the two-terminal measurements between the contacts? Are they linear?

5. The nonlinear Hall susceptibility extracted from the 4nm device is several times larger than that extracted from the 9nm device. Does this suggest that the observed nonlinear Hall effect (or skew scattering) might be a surface effect? Why does the 9nm device shows a much stronger temperature dependency in the linear conductivity?

6. Why does the 4nm device exhibit the skew scattering term and the side jump term having the same sign, while they are of opposite signs in the 9nm device?

7. Can the author present Fig. 4c in a larger temperature range down to 0K? From my rough estimation, the extrapolation of the $\epsilon_R=5$ meV curve would not intersect at 1 on the y-axis.

8. According to Fig. 3a, the nonlinear Hall voltage is almost zero under zero bias. Why in the wireless RF measurement, strong Hall signal up to mV can be observed? Are there any transport data of this 22nm device?

9. Since wireless RF rectification can occur without external bias, is it possible that bulk BiTeBr crystal could exhibit a similar behavior?

10. In Fig. 4a, the deviation from the quadratic scaling behavior is so significant. The authors attribute it to the influence of disorders. How do the disorders cause these drastic deviations? The authors should give more detailed discussions here.

11. Given the authors have theoretically derived the temperature dependence of the nonlinear Hall conductivity (eq. 5), the authors should derive the experimental nonlinear Hall conductivity from the nonlinear Hall susceptibility and fit it with eq. 5 to see if eq.5 can quantitatively describe the role played by the Rashba splitting in the skew scattering induced nonlinear Hall effect.

There are also minor issues:

1. The abstract includes a typographical error where "skew scattering" is misprinted as "screw scattering".

2. Could the authors clarify the unit of the color bar in Fig. 3a?

3. On page 4, the authors made the statement "Fig. 2e displays the plot of the nonlinear Hall susceptibility versus the conductivity of the device at all temperatures". This is inconsistent with data shown in Fig. 2e, which shows only the temperature dependence of the nonlinear Hall susceptibility.

Point-by-point response

We sincerely express our gratitude to all the reviewers for dedicating their time to thoroughly evaluating our work. We would like to emphasize that the comments and questions raised by the reviewers have helped us improved the paper. In response, we have diligently addressed each question and comment, augmenting our work with additional measurements and refining the clarity of our presentation. Below, we present a comprehensive, point-by-point response.

REVIEWER COMMENTS

Reviewer #1:

‘The paper by Lu et al. presents experimental studies of the second-order transport on 2D BiTeBr Rashba systems. Recently, the second order transport of noncentrosymmetric materials have attracted significant interest. The results present in this paper are definitely of interest. However, before I can support publication, the following issues need to be resolved:’

We extend our sincere gratitude to the reviewer for his/her insightful comments and accurate summary of our work. We are encouraged by the reviewer’s positive perspective that “*the results present in this paper are definitely of interest*”. We would like to emphasize that our study demonstrates a room temperature second-order nonlinear response in the Rashba material BiTeBr. The significant Rashba spin splitting in BiTeBr ensures the room temperature nonlinear response, which is useful for wireless RF rectification. Our research provides a straightforward approach for selection of materials exhibiting a substantial nonlinear response that endures above room temperature ($E_R > K_B T$). In the subsequent sections, we address the technical comments raised by the reviewer in a point-by-point manner.

Comment #1

1. The authors showed that the transverse nonlinear voltage V_{yxx}^2w is nonzero (i.e., injecting current along x and measuring the nonlinear voltage along y). Can the authors do the same experiments but rotate by 90 deg? V_{xyy} should be zero whereas V_{yyy} should be nonzero (with opposite sign). This is important to rule out various extrinsic second order effects (e.g. schottky diodes at the sample/contact junction, etc).

Response #1

We sincerely thank the reviewer for suggesting the measurement of the second-order nonlinear response while rotating the current direction by 90 degrees from the crystal axis. It is

crucial to rule out the extrinsic second order effects. To obtain this data, we fabricated disc device with 12 radially distributed electrodes. A harmonic current I^ω was injected through two of 12 electrodes and measured the voltage at first and second harmonic frequencies in both longitudinal and transverse directions. In our experiment, the x-axis is defined as the current direction and the y-axis denotes the transverse direction to the current. The measurement was conducted at $T = 2\text{K}$.

Figure R1. The second order nonlinear response measured at two orthogonal directions. a. Schematic image of the nonlinear response measured with applying current along the crystal axis. Scale bar, 10 μm . b, c. First and second harmonic longitudinal and transverse voltage as a function of alternating current I^ω when I^ω aligned along the crystal axis. d, e, f. Schematic image and the corresponding first and second harmonic response when rotating current direction by 90° .

Figure R1a-c illustrates the schematic and corresponding responses at first and second harmonic frequencies when the current is aligned along the crystal axis. In Figure R1b, first harmonic longitudinal voltage (V_x^ω) exhibits a linear increase with I^ω , while the transverse voltage (V_y^ω) remains minimal, indicating good ohmic contact and negligible electrode misalignment. In Figure R1c, the second harmonic transverse voltage ($V_y^{2\omega}$) scales linearly with the square of V_x^ω , while the longitudinal voltage ($V_x^{2\omega}$) shows negligible and irregular response. When we rotate the direction of I^ω by 90° , as depicted in Figure R1d-f, the $V_x^{2\omega}$ scales linearly

with the square of V_x with a negative sign, while $V_y^{2\omega}$ displays small and negligible response. The value of $V_y^{2\omega}$ and $V_x^{2\omega}$ measured at two orthogonal directions is almost equally.

These results agree with the reviewer's insightful suggestion that if a nonzero transverse nonlinear voltage $V_{yxx}^{2\omega}$ is observed with current along x direction, then a zero $V_{xyy}^{2\omega}$ and nonzero $V_{yyy}^{2\omega}$ with opposite sign can be measured when rotate the current direction by 90 degrees (Here x is the direction aligned to the in-plane crystal axis, y is the direction perpendicular to crystal axis). Consequently, our results effectively rule out extrinsic second order effects, such as Schottky diodes at the sample/contact junction.

In addressing concerns regarding the origin of extrinsic second-order effects, we have incorporated a discussion on symmetry-dependent nonlinear responses in the revised manuscript and included corresponding data in the supplementary information.

2. About the wireless rectification, Fig. 5a presents a device schematic that is cross-shaped, which is clearly different from the actual device picture in Fig. 1e. Are the authors using the same device shown in Fig. 1e to do wireless rectification or different device? More importantly, how do the authors control the electric field of the microwave and how do the authors know that the observed effect arises from a transverse second order response? In Ref. 18, the authors used antenna to control the direction of the incident microwave electric field. That would be useful if the authors want to claim that the rectification arises from the transverse nonlinearity.

Response #2

The device used for wireless RF rectification is a 22nm thick disc device with 12 radially distributed electrodes, as illustrated in Figure R2a. Considering the potential interference from the top layer graphite and hBN (utilized for the top electrical gate and safeguarding the thin BiTeBr flake) in the rectification process, we fabricated a device consisted solely of BiTeBr flake protected with PMMA polymer film. The wireless RF rectification data shown in Fig. 5 are all measured in this device. We acknowledge and apologize for any confusion caused by the schematic device structure in Fig. 5a. To provide a clearer depiction, we have made necessary modifications to Fig. 5a.

The reviewer raised an important query regarding the control of the microwave's electric field to ensure that the observed rectification DC voltage is a result of the transverse second-order response. We appreciate this insightful question and are thankful for the valuable suggestion to employ an antenna, a technique previously utilized by Kumar, D. *et al* in their wireless RF rectification paper on TaIrTe₄. In our study, we incorporated an antenna capable of transmitting polarized electromagnetic waves to measure the wireless RF rectification data. As depicted in Figure R2a, the dashed line indicates the crystal axis. We measured the rectified

DC voltage at the direction perpendicular to the crystal axis while rotating the incident electromagnetic wave's direction. The rectified DC voltage, presented in Figure R2b, as a function of RF source power, do not exhibit a clear, symmetry-dependent behaviour.

We attribute this lack of clear symmetry to two primary reasons. Firstly, unlike injecting current through electrodes, although we have used the antenna similar to the author of Ref.18 to control direction of the incident microwave electric field, the HPBW of our antenna (Half power beam width, characterizing the angle of microwave electric field in which relative power is more than 50% of the peak power) is about 20-25°. Thus, considering the in-plane three-fold rotational symmetry of BiTeBr, it is extremely challenging in fixing the incident electromagnetic wave in one direction without influencing the nearby 30° direction. Secondly, the second-order nonlinear response generated by BiTeBr obeys C_{3v} symmetry, e.g. the absolute peak value $|V_y^{2\omega}(0^\circ)| = |V_x^{2\omega}(30^\circ)|$. Thus, little deviation of the incident RF electromagnetic wave will lead to the rectified DC voltage contributed from mixing of both longitudinal and transverse direction. When considering the antenna beam width (20-25°), this kind of mixing is inevitable. These factors make it challenging to obtain precise angle-dependent wireless RF rectification data in BiTeBr.

We acknowledge that it would be inaccurate to assert that the rectification solely arises from transverse nonlinear response. We have revised Fig. 5a and revised the statement attributing rectification to nonlinear Hall effect in the revised manuscript.

Figure R2. a. The optical image of 22 nm BiTeBr device used for wireless RF rectification measurement. The dashed line indicates the crystal axis. Black line with arrows indicates the incident electromagnetic wave direction. θ is the incident angle of electromagnetic wave. Scale bar, 10 μm . b. The rectified DC voltage as a function of incident RF source power at different incident angle.

3. Interpretation of the data: In my opinion, what the authors observed is a second-order transverse signal but NOT a second-order Hall signal. Specifically, a Hall effect should be the antisymmetric component of the transverse signal, meaning that $\sigma_{yxx} = -\sigma_{xyx}$. By contrast, as the authors correctly pointed out in their own paper, the C_{3v} group of BiTeBr requires their nonlinear effect to be symmetric, i.e., $\sigma_{yxx} = \sigma_{xyx}$. In fact, the authors cited the theory work of Ref. 1 to support their experimental data (correctly so because Ref. 1 discussed the symmetric nonlinear effect in C_3 -symmetric systems, just like this experimental work). However, Ref. 1 has NEVER claimed that what they predicted is a Hall effect (they only call it a second order nonlinear effect). Therefore, I think the authors should change their claim from nonlinear Hall effect to nonlinear transverse transport. I recommend the authors to read the theory paper by Tsirkin and Souza (<https://arxiv.org/pdf/2106.06522.pdf>), which has a great pedagogical description of what can be interpreted as a Hall effect (being transverse is necessary but insufficient).

Response #3

We appreciate the reviewer for highlighting this important question. We acknowledge the existing ambiguity within the scientific community concerning the definition of the nonlinear Hall effect. In some references (*Phys. Rev. B* **100**, 195117 (2019), *Phys. Rev. Lett.* **127**, 277202 (2021), *Phys. Rev. B* **107**, 115142 (2023), *Science* **381**, 181-186 (2023)), it is defined as the antisymmetric component of the nonlinear conductivity tensor. However, in other source (*Nat. Nanotechnol.* **16**, 869-873 (2021), *Nat. Rev. Phys.* **3**, 744-752 (2021), *Phys. Rev. B* **105**, 045118 (2022)], it is defined as the transverse voltage response. We completely agree with the reviewer that it is more appropriate to define the nonlinear Hall effect as the antisymmetric part of the nonlinear conductivity tensor, in which $\sigma_{xyx} = -\sigma_{yxx}$ (<https://arxiv.org/pdf/2106.06522.pdf>). This definition ensures consistency with the linear Hall effect. Additionally, thanks to the reviewer's first comment, we observed both the second order longitudinal and transverse voltage $V_x^{2\omega}$ and $V_y^{2\omega}$ at two orthorhombic directions, respectively. The value of the measured $V_x^{2\omega}$ and $V_y^{2\omega}$ complies with the C_{3v} point group symmetry of BiTeBr that the second order conductivity $\sigma_{xxy} = \sigma_{xyx} = \sigma_{yxx} = -\sigma_{yyy}$ (*Sci. Adv.* **6**, eaay2497 (2020)). Therefore, we change our claim from nonlinear Hall effect to nonlinear transport response in our title and manuscript. Thanks again for the reviewer's suggestion.

Reviewer #3 (Remarks to the Author):

The nonlinear Hall effect is expected to be a new route for high-frequency energy harvesting. However, the lower working temperature of most previously reported nonlinear Hall materials

significantly limits its applications. In this manuscript, X. F. Lu et al. reported a room temperature nonlinear Hall effect in few-layer BiTeBr. A notable conclusion from their scaling analysis is that skew scatterings are the primary mechanism of the observed nonlinear Hall effect, which has traditionally been observed only at low temperatures. According to their theoretical analyses, the robustness of the nonlinear Hall effect in their devices is due to the strong Rashba spin-orbit coupling. They also demonstrated harvesting of high-frequency and low-power ambient electromagnetic energy with their prototype devices.

I believe their results are interesting and important and can be considered for publication in Nature Communications. However, there are still certain concerns that need to be addressed prior to the acceptance of this manuscript.

We sincerely thank the reviewer for the insightful comments and comprehensive assessment of our research. We are particularly encouraged by the reviewer's remarks that "*their results are interesting and important*". Our study demonstrates robust room temperature second order nonlinear response in Rashba material BiTeBr, which originates from skew scattering and holds promising application for harvesting high-frequency and low-power ambient electromagnetic energy. We believe that our work offers an important clue towards the pursuit of materials with substantial room-temperature nonlinear effect, which hold great potential for practical applications.

1. As mentioned in the manuscript, under C_{3v} symmetry, σ_{yxx} could be nonzero, while σ_{xyy} should always be zero. Can the authors demonstrate this concept with their existing devices?

Response #1

We sincerely thank the reviewer for raising this significant question regarding the nonlinear response in longitudinal and transverse directions under C_{3v} symmetry. This is helpful for us in verifying that the observed second order nonlinear electrical response in BiTeBr originates from skew scattering and adhering to the three-fold rotational symmetry of BiTeBr. Under C_{3v} symmetry, $\sigma_{yxx} = -\sigma_{yyy}$ is nonzero, and $\sigma_{xyy} = -\sigma_{xxx} = 0$, here x aligns along with the crystal axis and y is the direction perpendicular to crystal axis.

To comprehensively address the reviewer's inquiry and unravel the symmetry-dependent characteristics of nonlinear response in BiTeBr, we designed a disc-shaped device with 12 radially distributed electrodes precisely aligned with the crystal axis, as illustrated in Figure R2a. The black dashed line in the figure denotes the in-plane crystal axis of BiTeBr. A harmonic current I^{ω} was applied with an injection angle θ , and we measured the voltage at first and second

harmonic frequencies in both longitudinal and transverse directions. In our experiment, the x -axis is defined as the current direction and the y -axis denotes the transverse direction to the current. For all injection angles, the fundamental harmonic transverse voltage V_y^ω keep negligible compared to V_x^ω , indicating good alignment of the electrodes. Figure R2b shows the second-order nonlinear longitudinal and transverse responses as we rotated the current injection angle θ . Notably, both the longitudinal and transverse nonlinear response exhibit a three-fold symmetry, complying with the expected C_{3v} symmetry.

To establish that the observed nonlinear response complies with the threefold rotational symmetry of BiTeBr and is indeed arising from skew scattering, we have incorporated a discussion on symmetry-dependent nonlinear response in the revise manuscript and the symmetry-dependent data in supplementary information.

Figure R3. The angle dependence of the second order nonlinear response of BiTeBr in both longitudinal and transverse directions. a. Optical image of the 25nm-thick BiTeBr device. The electrode is deliberately aligned with the crystal axis, as marked with the dashed line. Scale bar, 10 μm . b. The second order nonlinear longitudinal and transverse voltages as a function of current injection angle θ .

2. As mentioned in the manuscript, for the C_{3v} symmetry, $\sigma_{yyy} = -\sigma_{yxx}$, the longitudinal second harmonic response should be observed when the current is applied along the y direction (perpendicular to the crystal axis). Such longitudinal second-order responses have already been reported in hBN/graphene moiré superlattice (ref. 32) and MnBi₂Te₄. Could the authors illustrate this in their experiment?

Response #2

We express our sincere gratitude to the reviewer for the insightful suggestion to conduct symmetry experiments, which really help us to improve the quality of our work. As stated in Response #1, under C_{3v} symmetry, $\sigma_{yxx} = -\sigma_{yyy}$ is nonzero, and $\sigma_{xyy} = -\sigma_{xxx} = 0$. This implies that the longitudinal nonlinear response should be observed when the current is injected perpendicular to the crystal axis. Our experimental data, presented in Figure R3b, show a clear three-fold symmetry for both longitudinal and transverse nonlinear response, complying with the C_{3v} symmetry of BiTeBr.

3. Can the authors show the data measured between the other pairs of contacts in the 4nm and 9nm device? Are they in agreement with the demonstrated data?

Response #3

Figure R4. Nonlinear response measured across different pairs of contacts. a. Schematic illustration of BiTeBr Hall bar device and the number of electrodes. b, c, and d, Second order longitudinal ($V_x^{2\omega}$) and transverse ($V_y^{2\omega}$) as a function of current measured across different pairs of contacts at 20K, 30K, and 40K.

We would like to express our gratitude to the reviewer for raising this comment, which is helpful to enhance the quality of our work. It is crucial that the second-order nonlinear response arising from skew scattering remains consistent when measured across different pairs of contacts. In Figure R4, we show the nonlinear response ($V_x^{2\omega}$ and $V_y^{2\omega}$) measured between various pairs of contacts in 4 nm device. These results clearly demonstrate the second-order nonlinear response does not rely on the specific pairs of contacts used for measurement. This finding further excludes any extrinsic second-order effects (e.g. accidental diodes between the sample and contact junction).

We have incorporated the data of nonlinear response measured across different pairs of contacts in the supplementary information to further verify that the observed nonlinear response is not attributed to any extrinsic effect.

4. What's the current-voltage characteristic of the two-terminal measurements between the contacts? Are they linear?

Response #4

We sincerely thank the reviewer for raising this question. We have measured all devices' two-terminal current-voltage (I - V) characteristics before nonlinear response measurement. The summarized results are presented below. Figure R5 shows the two-terminal I - V curves measured in 4 nm-thick BiTeBr device, revealing a consistently linear behaviour across all electrodes, suggesting good ohmic contact. Furthermore, the two-terminal I - V curves for the 9 nm-thick and 25 nm-thick disk devices are also presented in Figure R6, which all exhibit linear behaviour and confirm the good ohmic contact across all electrodes.

We have incorporated these two-terminal I - V curves measured across different contacts in supplementary information to demonstrate the good ohmic contact of our devices.

Figure R5. Two-terminal DC characteristics of 4 nm-thick BiTeBr device. **a**, Schematic illustration of 4 nm-thick BiTeBr Hall bar device with labelled electrode numbers. **b** to **f**, Two-terminal I - V characteristics for all electrodes.

Figure R6. Two-terminal DC characteristics of 9 nm, 15 nm, and 25 nm thick BiTeBr devices, respectively.

5. The nonlinear Hall susceptibility extracted from the 4 nm device is several times larger than that extracted from the 9 nm device. Does this suggest that the observed nonlinear Hall effect (or skew scattering) might be a surface effect? Why does the 9 nm device shows a much stronger temperature dependency in the linear conductivity?

Response #5

We appreciate the insightful observation posed by the reviewer. We agree with the reviewer that the observed difference in nonlinear response between the 4nm and 9nm devices suggests a notable influence of surface effects, particularly in the case of the thinner 4nm device. Notably, for BiTeBr, previous studies have demonstrated that the Rashba spin-splitting is more pronounced at the surface (*Phys. Rev. Lett.* **108**, 246802 (2012); *New J. Phys.* **15**, 075015 (2013); *EPL*, **116** 57003(2016)). Due to this enhanced surface Rashba spin-splitting, the skew scattering induced by the Rashba effect becomes more evident in thinner samples. The nonlinear response associated with surface states has been observed in other systems, like TaIrTe₄ (*Nat. Nanotech.* **16**, 421-425 (2021)) or Bi₂Se₃ (*Nat. Commun.* **12**, 698 (2021)).

Nevertheless, the contribution of bulk states to the nonlinear response in BiTeBr is still remarkable. As evidence, we fabricated an additional 15 nm BiTeBr device and present the corresponding data in Figure R7. The data clearly show an obvious nonlinear response, and the amplitude of the nonlinear susceptibility $E_y^{2\omega}/(E_x)^2$ is comparable to that of the 9 nm device. Crucially, the nonlinear susceptibility does not decrease with the increasing thickness of the device (as seen in TaIrTe₄ and Bi₂Se₃ nonlinear responses), suggesting that the nonlinear response in BiTeBr is not solely attributable to surface effects. The Rashba spin-splitting in the bulk state also plays a significant role in the observed nonlinear response, particularly in thicker devices.

Regarding the stronger temperature dependence of the linear conductivity in the 9nm device, this can be attributed to the smaller influence of surface states, leading to a behavior similar to bulk-like properties in the thicker device. In the 9nm device, phonon scattering might dominate, contributing to a more pronounced temperature dependency. Another contributing factor could be the increased disorder or defect introduced within the electrically active regions in thinner samples (*Nanomaterials* **11**, 2826 (2021); *Science* **324**, 1314 (2009)), which is indicated by the reduced residual resistance ratio (*RRR*) value ($RRR = (RRR(300\text{ K}) - R(0\text{ K}))/R(0\text{ K})$). Consequently, the thinner sample exhibits weaker temperature dependence.

In the revised manuscript, we have incorporated a discussion on surface spin-splitting in BiTeBr producing stronger skew scattering in thin device to provide a explanation for the observed larger nonlinear response in thin device.

Figure R7. Nonlinear transport response of BiTeBr device with thickness $\sim 15\text{nm}$. a, First-harmonic longitudinal and transverse voltage as a function of I^ω at 10K. b, $V_y^{2\omega}$ depends linearly on $(V_x)^2$ and changes sign when the current direction and voltage probe electrodes are simultaneously reversed. c, $E_y^{2\omega}$ dependent of $(E_x)^2$ measured at temperature ranging from 10 to 300K. d and e, The nonlinear susceptibility $E_y^{2\omega}/(E_x)^2$ and conductivity σ as a function of temperature, respectively. f, The nonlinear susceptibility $E_y^{2\omega}/(E_x)^2$ as a function of σ^2 .

6. Why does the 4nm device exhibit the skew scattering term and the side jump term having the same sign, while they are of opposite signs in the 9nm device?

Response #6

We sincerely thank the reviewer on the nice comments. In the below, we give a possible explanation.

For the 4nm device, due to the thickness, the increased surface-to-volume ratio might lead to a significant contribution from surface states or surface-related scattering mechanisms. These surface effects could modify the scattering landscape in such a way that skew scattering and side jump mechanisms yield contributions with the opposite sign. It is worth noting that in Bi_2Se_3 , where the nonlinear response originates from topological surface state, the skew scattering and side jump provides opposite contribution to the nonlinear response (*Nat. Commun.* **12**, 698 (2021)). In the 9nm device, the reduced influence of surface states might lead to a scenario where the bulk properties of the material, combined with its electronic structure,

determine the signs of the skew scattering and side jump contributions. The bulk spin-orbit interactions, band structure details, and scattering potentials might result in the positive sign for both skew scattering and side-jump mechanisms. We measured another 15 nm BiTeBr device, which shows same sign of the side jump and skew scattering contribution (See Figure.R7 f).

We also notice that in the anomalous Hall effect, the side jump contribution will change sign with different thickness (e.g. FeCo films grown on MgO (001) thin films, (*Phys. Rev. B* **107**, 094418 (2023))). The sign of side jump contribution changes from negative to positive with the increase of thickness, which resembles our finding and is attributed to the competing scattering contribution changes from dynamic-scattering dominated to static-scattering dominated.

Further studies, perhaps detailed theoretical modelling, might provide deeper insights into these observations.

We have added a discussion in the revised manuscript to give a possible explanation to the sign reversion of side jump with increasing the thickness of BiTeBr device.

7. Can the author present Fig. 4c in a larger temperature range down to 0K? From my rough estimation, the extrapolation of the $\epsilon_R=5$ meV curve would not intersect at 1 on the y-axis.

Response #7

We thank the reviewer for this good suggestion. Theoretically, we have derived that the temperature dependence of the second-order conductivity $\frac{\sigma_{yxx}^{(2)}(T)}{\sigma_{yxx}^{(2)}(0)} \simeq \left[\frac{\tau(T)}{\tau_0}\right]^3 \chi(\epsilon_R/T)$, where $\chi(\epsilon_R/T) = \tanh \frac{\epsilon_R}{2k_B T}$ is the coefficient associated with the Rashba SOC, as shown in eq.(5) of the main text. The Rashba coefficient is plotted as a function of temperature in Figure R8a. The coefficient remains close to unity when $k_B T \ll \epsilon_R$, and it starts to decay when $k_B T$ becomes comparable with ϵ_R .

Next, we estimate the temperature dependence of the scattering time τ according to the conductivity as shown in fig. 2f in the main text. The resistance shows a linear dependence on temperature when $T > 20$ K, which allows us to approximate the scattering time as $\tau^{-1} = \tau_0^{-1}(1 + aT)$, with $a \approx 1.1 \times 10^{-3} \text{ K}^{-1}$ obtained from fitting our experimental data in Figure 2f. Taking the scattering time into consideration, we obtain the temperature dependence of the second-order conductivity $\sigma_{yxx}^{(2)}(T)$, as depicted in Figure R8b. We use dashed lines for $T < 20$ K, because the linear dependence of the resistance (inversed scattering time τ^{-1}) on temperature is only observed for $T > 20$ K in the experiment.

In the revised manuscript, we have extended the temperature range down to 0 K as the referee suggests for clearer clarification.

Figure.R8 **a**, and **b**, The calculated Rashba coefficient $\chi(\epsilon_R/T)$ and second-order conductivity $\sigma_{yxx}^{(2)}(T)$ as a function of temperature for Rashba band splitting $\epsilon_R = 42$ meV (pink) and $\epsilon_R = 5$ meV (cyan).

8. According to Fig. 3a, the nonlinear Hall voltage is almost zero under zero bias. Why in the wireless RF measurement, strong Hall signal up to mV can be observed? Are there any transport data of this 22nm device?

Response #8

We are grateful to the reviewer for these valuable comments. In Fig. 3a, when the gate voltage is zero, the nonlinear transverse response is almost zero with 2 μ A input current, even though its actual value should be a few μ V. This might give an impression of a zero-value due to the scale of the figure,

The wireless RF measurement was conducted on 22nm device at room temperature under zero gate voltage. Figure R9 shows the second order nonlinear transverse response measured at 22nm device at room temperature (also shown in supplementary Figure S3), with a value of ~ 2 μ V under 2 μ A input current. In wireless RF measurement, the incident electromagnetic wave absorbed by the BiTeBr device acts as the input AC current. Given that the applied RF source power reaches up to 40 mW, this power is much larger than the input power used in transport measurement (given the resistance of 22nm BiTeBr device at room temperature is ~ 1000 Ω). Therefore, a rectified DC voltage up to mV can be observed with such a large RF power.

Figure R9. Nonlinear transverse response measured in 22nm device at 300K.

9. Since wireless RF rectification can occur without external bias, is it possible that bulk BiTeBr crystal could exhibit a similar behavior?

Response #9

We are grateful to the reviewer for this valuable comment. Because bulk inversion symmetry is broken in the crystal, the bulk crystal should have Rashba-Dresselhaus type band splitting. As stated in Response #5, thicker BiTeBr devices (those exceeding 9nm) will also have Rashba-Dresselhaus type band splitting, and we have observed the nonlinear response in BiTeBr device with thickness 15 nm, 22 nm, and 25 nm under zero gate voltage. Additionally, the wireless RF rectification data shown in Fig. 5 were measured on 22 nm thick BiTeBr device without external bias. In our perspective, the device with thickness up to 22 nm and 25 nm can be regarded as bulk crystal. Therefore, the bulk BiTeBr crystal should exhibit similar rectification behavior.

10. In Fig. 4a, the deviation from the quadratic scaling behavior is so significant. The authors attribute it to the influence of disorders. How do the disorders cause these drastic deviations? The authors should give more detailed discussions here.

Response #10

We sincerely thank the reviewer for the insightful observation. As the reviewer pointed out, the quadratic fitting of the nonlinear conductivity significantly diverges at low temperature for the 4 nm BiTeBr device, while this deviation is less pronounced for the 9 nm device. Our hypothesis attributes this observed behavior to a pronounced increase in disorder or impurities within the thinner device. The increased disorder or impurities in thinner device (4 nm) can be evidenced by the observation of localization behavior from the downturn of conductance at low

temperature, as depicted in Fig. 2f. Such a trend is common in 2D thin flakes (e.g. *Nat. Mater.* **17**, 778–782 (2018)). Moreover, research indicates that the increased disorder or impurities could largely enhance the nonlinear conductance, especially when the Fermi-level is located within the band (Arxiv: 2309.07000 (2023)). Therefore, as the temperature lower, the effects of disorder or impurity scattering become increasingly dominant, leading to a significant deviation from the expected quadratic scaling behavior.

We expect that further investigation on the impact of disorder and impurities on the nonlinear response can provide further insights on this.

11. Given the authors have theoretically derived the temperature dependence of the nonlinear Hall conductivity (eq. 5), the authors should derive the experimental nonlinear Hall conductivity from the nonlinear Hall susceptibility and fit it with eq. 5 to see if eq.5 can quantitatively describe the role played by the Rashba splitting in the skew scattering induced nonlinear Hall effect.

Response #11

We completely agree with the reviewer that it is important to verify whether the equation derived from theory can quantitatively matching with our experimental data. Since $\frac{E_y^{(2)}}{(E_x)^2} = \frac{\sigma_{yxx}^{(2)}}{\sigma}$, the second order nonlinear conductivity ($\sigma_{yxx}^{(2)}$) can be obtained from the experimental nonlinear susceptibility $\frac{E_y^{(2)}}{(E_x)^2}$ and conductivity σ . The nonlinear conductivity as a function of temperature for different thickness BiTeBr devices is shown in Figure R10a-c.

To fit with Eq. (5) in the main text, a constant C_0 was brought in for ease of fitting, $\frac{\sigma_{yxx}^{(2)}(T)}{\sigma_{yxx}^{(2)}(0)} = C_0(1 + aT)^{-3} \tanh \frac{\epsilon_R}{2k_B T}$. For the 4 nm device, we selected $\sigma_{yxx}^{(2)}(0)$ as 0.22 SV^{-1} to determine $\sigma_{yxx}^{(2)}(T)/\sigma_{yxx}^{(2)}(0)$. This ratio, as a function of temperature for the 4 nm device, is illustrated in Figure R10d with red scattered circles. By fitting the data, we obtain $a = 3.5 \times 10^{-3} \pm 5 \times 10^{-3} \text{ K}^{-1}$, $\epsilon_R = 0.05 \pm 0.009 \text{ eV}$, $C_0 = 1.04 \pm 0.1$, with the fitting curve presented with red solid line in Figure R10d. The experimental data $\sigma_{yxx}^{(2)}(T)/\sigma_{yxx}^{(2)}(0)$ fits well with the theory Eq. (5), implying it can indeed provide a quantitative description of the Rashba splitting's role in skew scattering-induced nonlinear response. The slight discrepancy between the fitted ϵ_R (50eV) and the Rashba splitting energy in BiTeBr (42 eV) might arise surface effects contributing to the nonlinear response. Furthermore, the larger fitted value of 'a' ($3.5 \times 10^{-3} \text{ K}^{-1}$) compared to the one derived from the temperature dependence of conductivity ($a \sim 1.1 \times 10^{-3} \text{ K}^{-1}$) may be attributed to influences of surface state and device disorder.

Adopting with the same method, the experimental data $\sigma_{yxx}^{(2)}(T)/\sigma_{yxx}^{(2)}(0)$ and its fitting curve with Eq. (5) for 9 nm and 15 nm thick devices are shown in Figure R10e, f. The fitted value of ‘a’ increase with thickness, while the ϵ_R decrease, indicating the influence of the surface effect is reducing.

We have incorporated the content of fitting the experimental second-order conductivity ($\sigma_{yxx}^{(2)}$) to Eq. (5) in the revised manuscript to show the good agreement between theoretical equation and experimental data. The corresponding figures are included in supplementary information.

Figure R10. The experimental second order nonlinear conductivity ($\sigma_{yxx}^{(2)}$) and its corresponding theoretical fitting for different thickness BiTeBr device. a and d, Experimental $\sigma_{yxx}^{(2)}$ in 4 nm device and its corresponding fitting of $\sigma_{yxx}^{(2)}(T)/\sigma_{yxx}^{(2)}(0)$ using eq. (5) in the main text. b and e, Experimental $\sigma_{yxx}^{(2)}$ in 9 nm device and its corresponding theoretical fitting of $\sigma_{yxx}^{(2)}(T)/\sigma_{yxx}^{(2)}(0)$. c and f, Experimental $\sigma_{yxx}^{(2)}$ in 15 nm device and its corresponding theoretical fitting of $\sigma_{yxx}^{(2)}(T)/\sigma_{yxx}^{(2)}(0)$.

There are also minor issues:

1. The abstract includes a typographical error where "skew scattering" is misprinted as "screw scattering".

Response #1

We sincerely apologize for the typographical error in the abstract. We have corrected this typo in the revised manuscript. Thank you for pointing it out.

2. Could the authors clarify the unit of the color bar in Fig. 3a?

Response #2

We apologize for the negligence regarding the missing unit of the color bar in Fig. 3a. The unit of the color bar is μV . This has been added to Fig. 3a in the revised manuscript. Thank you for bringing this to our attention.

3. On page 4, the authors made the statement "Fig. 2e displays the plot of the nonlinear Hall susceptibility versus the conductivity of the device at all temperatures". This is inconsistent with data shown in Fig. 2e, which shows only the temperature dependence of the nonlinear Hall susceptibility.

Response #3

We sincerely apologize for the confusion. We have revised this statement to '*The temperature dependence of $E_y^{2\omega} / (E_x)^2$ and conductivity (σ) of the device are shown in Fig. 2e and 2f, both of which exhibit similar behaviour.*' in the revised manuscript.

Reviewers' Comments:

Reviewer #1:

Remarks to the Author:

I appreciate the authors' careful and detailed replies. The authors have satisfactorily addressed all my comments. I recommend publication.

Reviewer #3:

Remarks to the Author:

I appreciate that the authors made significant efforts to address the questions raised in my first report. The response to each of my questions is convincing. In particular, the new experimental data measured on the disc-like devices answered the questions which I am most concerned about. I now recommend it for publication in Nature Communications.

One minor suggestion:

I noted ref.[20] has been updated in arXive. The new version reports that the strong nonlinear Hall response previously observed in the NbP device is indeed not from NbP, but from the Pt electrodes deposited by focused ion beam. I suggest that NbP in table 1 is replaced by "focused ion beam deposited Pt".

Point-by-point response

We extend our gratitude to all the reviewers for generously investing their time in thoroughly evaluating our work. We are glad that both reviewers have recommended publication of our work. In response, we have addressed the comment raised by reviewer #3. Below, we provide a detailed, point-by-point response.

REVIEWER COMMENTS

Reviewer #1

I appreciate the authors' careful and detailed replies. The authors have satisfactorily addressed all my comments. I recommend publication.

We sincerely appreciate the reviewer's positive assessment and the recommendation for publication of our work. His/her constructive comments have significantly contributed to enhancing the quality of our manuscript. Thanks once again for your valuable time and feedback.

Reviewer #3

I appreciate that the authors made significant efforts to address the questions raised in my first report. The response to each of my questions is convincing. In particular, the new experimental data measured on the disc-like devices answered the questions which I am most concerned about. I now recommend it for publication in Nature Communications.

We sincerely thank the reviewer for recommending the publication of our work. We are especially encouraged by your comment that *"The response to each of my questions is convincing."* Thank you again for your nice suggestions and strong support.

One minor suggestion:

I noted ref.[20] has been updated in arXiv. The new version reports that the strong nonlinear Hall response previously observed in the NbP device is indeed not from NbP, but from the Pt electrodes deposited by focused ion beam. I suggest that NbP in table 1 is replaced by "focused ion beam deposited Pt".

Response

We extend our sincere gratitude to the reviewer for providing this valuable suggestion. We apologize for the oversight regarding the updated version of ref. [20] in arXiv. Following the reviewer's guidance, we have corrected the "NbP" in Table 1 to "focused ion beam deposited Pt". We are grateful for the reviewer's professional and meticulous assessment of our work, which has significantly enhanced its quality. Thank you once again for your valuable contributions.